# Investigating the conception of collaborative learning (CL) and student engagement in the acquisition of practical skills (SEPSA) among prospective physical education and sports students

**Yerlan Temirkhanov** [ORCID]**\*, Taiyrzhan Iskakov, Mira Iralina, Aidyn Zhumagulov, Gulnaz Atagulova, Saltanat Boztayeva**

Abai Kazakh National Pedagogical University, Almaty, Kazakhstan

\* Erlan_1993kz@mail.ru

## Abstract

The study aims to explore the association between collaborative learning and practical skills acquisition (SEPSA) among 310 students from second-year, third-year, and fourth-year (First stage of higher education) from the Institute of Arts, Culture, and Sports- Abai Kazakh National Pedagogical University. The data was collected using the time-lag approach at three intervals; 3rd week, 7th week, and 14th week. The mediation analysis suggests that collaborative learning (CL) has a positive mediating association with self-efficacy, and student engagement in practical skills acquisition (SEPSA). Additionally, collaborative learning (CL) has a positive mediating association with value-benefits, and practical skills acquisition (SEPSA). Furthermore, Collaborative learning (CL) has a positive significant association with practical skills acquisition (SEPSA). Our findings highlight the important potential of CL for increasing SEPSA. The finding of the study has implications for higher education teachers, students, administrators, and policymakers for developing more effective teaching and learning approaches using the concept of sharing and discussion with a specific focus on students' engagement.

## 1. Introduction

The collaborative learning (CL) approach of learning in higher education is widely approved pedagogy specifically within the school of thought that values peer-to-peer learning. Dillenbourg [1] defines it as, "a situation in which two or more people learn or attempt to learn something together" (p. 1) Yang [2]. Through CL students generate innovative knowledge and value their work [3–5]. Educational institutes are rapidly integrating innovative methodologies to support teaching approaches. The transformation from traditional teaching approaches to digital learning experiences has been reshaping the learning experiences [6, 7]). Advancement in the technologies that aid the learning experience has boosted student engagement, active

**Data Availability Statement:** All relevant data are within the manuscript and its Supporting information files.

**Funding:** The author(s) received no specific funding for this work.

**Competing interests:** NO authors have competing interests.

participation, and engagement in collaborative learning [8]. In recent years, many researches were carried out to highlight the importance and effectiveness of collaborative learning [9–11];). Collaborative learning promotes self-efficacy, motivation, and results in better learning outcomes [12] Traditional teaching methods are now being replaced with modern active learning approaches which are specifically designed to expedite student-centered approaches [13]. The active learning approach focuses on the importance of students' engagement in the learning process through discussion and problem-solving attitude [14]. One of the features of active learning is its ability to encourage critical thinking in the students [15, 16]). In an interactive learning environment, students act as a learning resource for each other thus building more innovative learning experiences [17]. Furthermore, student engagement in practical skills acquisition (SEPSA) has gained momentum in higher education institutes as through SEPSA students acquire practical skills that enhance their employability and career development probabilities [18, 19]. Collaborative learning (CL) plays an important role in practical skills acquisition as through collaborative learning students engage in group discussion and generate innovative solutions and learning methodologies [9, 13, 20]. SEPSA requires a deeper understanding of students' practical skills behavior and their motivation to pursue [13, 18, 21] Moreover, limited areas are explored between CL and SEPSA, thus arises the need to address the theoretical and empirical gap that needs to be explored scientifically. Additionally, understanding the internal dynamism will help to give insight into whether CL improves SEPSA. We opted for expectancy-value theory to fill this gap as this theory has the potential to offer a deeper understanding of CL and SEPSA via the psychological domain. EVT addresses two factors that impact students' motivation to learn that is, students' expectation regarding achievement and students' value linked with tasks/goals associated with achievement.

## 2. Theoretical framework

### 2.1 The integration of Expectancy-value theory (EVT) and Collaborative Learning (CL)

Fig 1, depicts the Expectancy-value theory (EVT) model that was developed by Atkinson in 1964 and was further developed by Eccles (1984) and applied in educational psychology by Eccles and Wigfield [22, 23] EVT explains the motivation of an individual regarding specific task completion to achieve a certain goal. Furthermore, motivation is highest when a person is challenged on a specific task and also a person believes that a certain goal is achievable. EVT explores a person's motivation based on expectancy beliefs and value beliefs. Expectancy beliefs are the extent to which a person feels he/she can complete the task successfully that include their beliefs and abilities. Whereas, value beliefs refer to the worth a person has assigned to a particular task such as enjoyment and usefulness linked with task completion. Lastly, cost beliefs can influence both value and expectancy beliefs. Cost belief consists of the efforts and time required to complete the task. In the current study, we have used EVT to explore the dimensions of CL and SEPSA. Moreover, our focus was to link EVT theory to explore the association between CL and SEPSA.

## 3. Hypotheses development

### 3.1 Collaborative Learning—CL and Student Practical Skills Acquisition–SEPSA as a framework for effective learning

Collaborative Learning Collaborative Learning (CL) significantly impact the learning experiences through developing skills to critically analyze the context [24] (Ma et al., 2018). Current study adopted the definition of CL from Johnson and Johnson [25] that explains CL as, set of

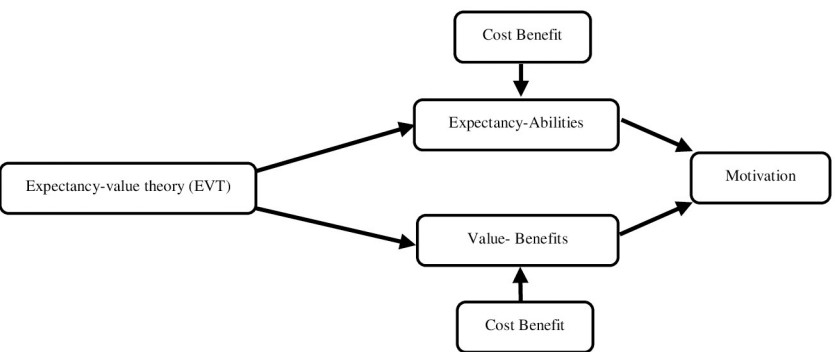

**Fig 1. Expectancy-value theory (EVT).**

teaching and learning strategies which allows students to work together to accomplish shared goals, seek outcomes that are beneficial to all, discuss materials with each other, help each other to understand concepts, and encourage each other to work hard. Moreover, literature also address the obstacles that impact effectiveness of CL, these obstacles unfair or unequal student's participation in discussion, deprived communication skills, introvert, and social background [13, 18] (Le et al., 2018; Okolie, 2022). Also, teachers experiences the obstacles while planning and organizing CL activities such as ineffective group tasks or improper time management for each task [13, 18]. Moreover, CL allows students to develop social interaction [26]. Social interaction allows students to build up their self-efficacy [27]. Previous studies on Self-efficacy have shown positive association with learning as students with high self-efficacy are more willing to put efforts in the learning and tasks especially when they are facing difficulties [28–32]. However, the role of self-efficacy in student engagement in practical skills acquisition (SEPSA) through collaborative learning needs to be explore in order to address this gap following hypotheses was formulated,

**Hypothesis H1**. Collaborative Learning (CL) has a significant positive associated with Self-efficacy

**Hypothesis H2**. Self-efficacy has a significant positive association with SEPSA

In active learning the most important factor is social interaction [33]. According to sociocultural theory of learning, social interaction results in innovation through ideas and knowledge generation thus motivating students to engage, share, and develop ideas [34] (Vuopala et al., 2016). Furthermore, interaction between students and with teachers results in students' engagement [35] Technology is playing important role in student engagement as the power of social media in facilitating learning through collaboration [36–38] However, with positive aspect the negative factors always exists in case of social media online toxicity i.e. cyberbullying and cyberstalking [39] Student engagement in practical skills acquisition (SEPSA) equipping students with set of skills that help students to gain employment and career advancement [18, 19]. Moreover, student engagement is directly associated with motivation [40, 41]. Collaborative learning allows students to innovative and when students are engaged in skills that can benefit them in real world they feel motivated and thus develop new approaches and develop new set of skills [18, 42, 43]. Student motivation in learning has been explored in various studies [44–46]. Furthermore, according to Expectancy-value theory (EVT) student compare efforts with the skills they are being taught and its usefulness. If the efforts are worthy they feel motivated [47]. However, the role of practical skill acquisition The current study took

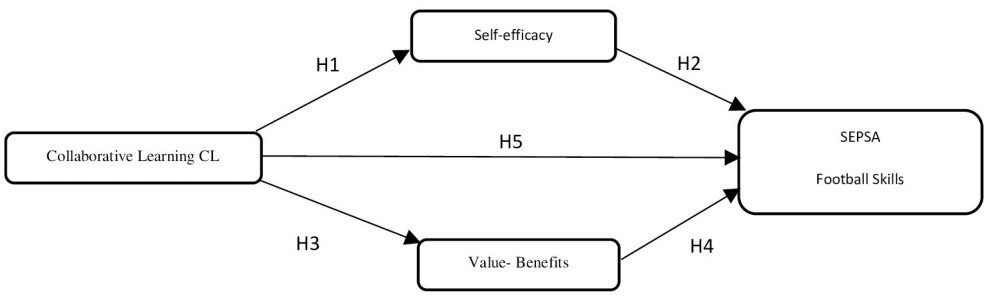

**Fig 2. Conceptual model.**

inspiration from EVT and linked CL with SEPSA with mediating effect of value-benefits therefore, the study crafted following hypotheses,

**Hypothesis H3**. Collaborative Learning (CL) has positive association with value-benefits.

**Hypothesis H4**. Value-benefits has positive association with SEPSA (Football Skills).

## 4. Hypotheses conceptual model

The study's conceptual model as presented in Fig 2, was crafted utilizing insights from existing literature and addressing the identified research gap. This model visually portrays how the research aims to bridge this gap through its proposed framework.

Fig 2 illustrates the study's conceptual model, depicting the five hypotheses concerning the variables under investigation.

## 5. Methods

### 5.1 Sample and procedure

In this study students were purposively selected from second-year, third-year, and fourth-year of their undergraduate program (first stage of higher education) in the faculty of arts, culture and sports from Abai Kazakh National Pedagogical University. Furthermore, students with specialization in physical culture and sport were purposively selected. In total 310 students were included in the study out of which second-year students were 60, third year students were 157, fourth-year students were 93. We obtained consents from all student participants and assured them anonymity and confidentiality. Fifteen teachers from the institute of arts, culture and sports- The National Pedagogical University were recruited for the development of CL framework in association with practical skills acquisition (SEPSA) based on the curricula. Students were divided into small groups each group with maximum of eleven students in a group and were assigned different CL tasks based on practical skills. Students were divided into two equal groups 155 students per group. Furthermore, these two groups were divided into 28 teams. These 28 teams were assigned different practical under collaborative learning (CL) approach tasks under the supervision of 28 head coaches. These tasks were divided into developing game plan: offensive strategies (Fly route, Slant route, Out route, Screen Pass, Button Hook, Corner Route, Hail Mary, and Seam route), and defensive strategies (Man-to-man, Coverage shells (cover 2, cover 3), Zone blitz, Tampa 2, 46 defense, and 5–5–1 Two-level defense). The CL activities were designed for 14 weeks (excluding summer breaks).

Group members were not allowed to switch groups and each group had assigned a unique identification code that is a combination of students' registration number. The data was collected with a time-lag approach. The first data was collected at the start of the semester (3[RD] Week) where students were in initial stages of CL and SEPSA, the second of data was collected at the mid of semester (7[th] week) where students were well aware of CL and SEPSA, whereas, the last stage of data collection was at the end of semester (14[th] week) where students fully experienced the concept of CL and SEPSA. The data collected at three different points were analyzed using SPSS v21 and Process v4.2 by Andrew F. Hayas (Model 4). The data was collected from 02 January, 2023 till 12 May, 2023.

## 5.2 Measure

All constructs and items were measured on a Likert five-point scale anchored from 1(strongly disagree) to 5(strongly agree).

**Collaborative Learning (CL).**   Self-developed scale consisting of 15 items was used to measure CL, the scale was developed using procedure and guidelines by Aithal & Aithal [48]. A sample item example: "I feel that the group task helps me to learn and explore new concepts as a team." The Cronbach's alpha value ($\alpha$ 0.92) which is well accepted range as per social science standards [49] (Baistaman, 2020).

**Self-Efficacy (SE).**   SE scale was adopted from the work of Schwarzer and Jerusalem [50] and was modified to fit the scope of the study. The scale consists of 10-items designed to measure students' confidence in solving problems, facing challenges and dealing with unexpected tasks. An example item is "I can always manage to solve difficult problems if I try hard enough." The Cronbach's Alpha value for this scale ($\alpha$ 0.90).

**Student Engagement in Practical Skills Acquisition (SEPSA).**   The scale for SEPSA was adopted from the work of Zhoc [51] and was modified to fit the scope of the study. The scale consists of 8-items. The sample item is "I frequently join other students to discuss the practical skill task." The Cronbach's alpha value for this scale ($\alpha$ 0.93).

**Value-Benefits (VB).**   The VB scale was self-developed consisting of 10-items. The sample item is "I believe that the time and effort I have invested in learning practical skills has been worthwhile." The Cronbach's alpha value for this scale ($\alpha$ 0.91).

## 6. Analytical procedures and results

Fig 3 depicts the association among the research variables in terms of beta values, standard error, and significance value.

## 6.1 Sobel test

The indirect and direct effect was measured using Process v4.2 by Andrew F. Hayas (Model 4) in SPSS v21 along with Sobel test. Table 1 shows the outcome of the process, mediating analysis was performed to explore CL association with SEPSA.

**6.1.1 Direct effect CL -SEPSA.**   The association between CL and SEPSA was statistically significant ($\beta$ = .017; p = 0.000). Therefore, we accept Hypothesis H5. According to Baron and Kenny [52] if the association is positive we can progress with the model however, according to Cho & Lee [53] even with non-significant association we can progress in model. In this study we have followed Baron and Kenny recommendation.

**6.1.2 Indirect effect CL -SE -SEPSA.**   The mediating analysis suggested that CL has a positive and statistical significant association with SE ($\beta$ = .950; p = 0.000). Moreover, SE has a statistically significant association with SEPSA ($\beta$ = .602; p = 0.000). The combine effect obtained using Process v4.2 was measure through a × b = .5710 and the corresponding Sober t-

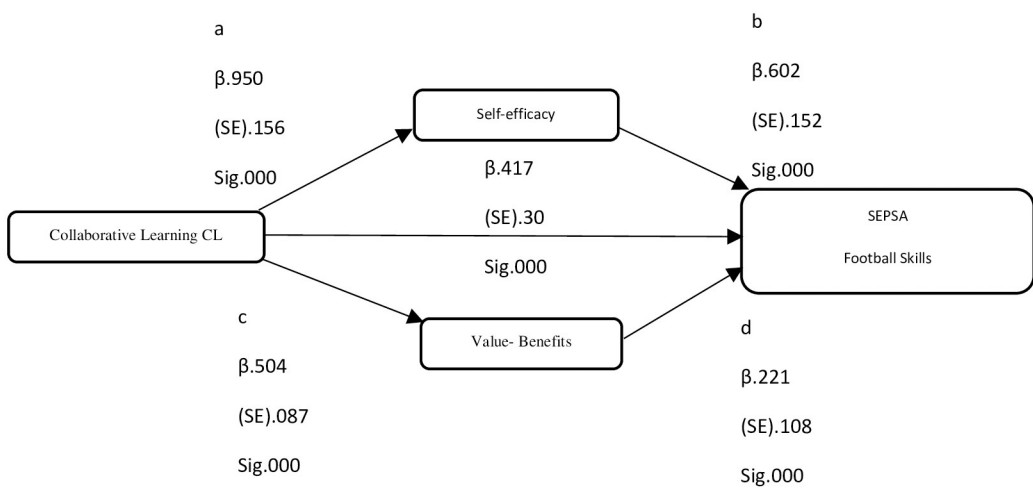

**Fig 3. Results.**

value = 3.320 with p = 0.000 therefore, we concluded there exists a positive association between CL and SEPSA via SE. Thus we accept Hypothesis H1 and Hypothesis H2.

**6.1.3 Indirect effect CL -VB -SEPSA.**　Similarly, the mediating analysis suggested that CL has a positive and statistical significant association with VB ($\beta$ = .504; p = 0.000). Moreover, VB has a statistically significant association with SEPSA ($\beta$ = .221; p = 0.000). The combine effect was measure through c × d = .111 and the corresponding Sober t-value = 1.929 with p = 0.050 therefore, we concluded there exists a positive association between CL and SEPSA via VB. Thus we accept Hypothesis H3 and Hypothesis H4.

Table 2 shows the bivariate correlations analysis, SE has a very strong positive and statistically significant (p ≤ 0.05) association with CL (r = .874). VB has strong positive association with CL (r = .617) and with SE (r = .691). SEPSA has a strong positive statistically significant correlation with CL (r = .787) and moderate positive correlation with SE (r = .580). Lastly, SEPSA has statistically insignificant correlation with VB.

## 7. Teams progress

In total 24 teams were formulated for the study each with maximum number of 11 participants, the teams progress was measured at three different intervals (3$^{rd}$ week, 7$^{th}$ week, and 14$^{th}$ week) along with students engagement with SEPSA.

**Table 1. Results showing all direct effects and indirect effects.**

| Paths | Estimate | Sobel test t-value | Std. Error | p-value |
|---|---|---|---|---|
| Indirect Effect | a×b | 3.32013157 | 0.17225221 | 0.000 |
| **CL -SE -SEPSA** | 0.5719 | | | |
| Indirect Effect | c×d | 1.92946284 | 0.05772798 | 0.050 |
| **CL -VB -SEPSA** | 0.111 | | | |
| Paths | Unstandardized Coefficients | Std. Error | Sig P-value | |
| Direct Effect | $\beta$ = .417 | .30 | .000 | |
| **CL -SEPSA** | | | | |
| **Model Summary** | R = .804 | Std. Error = 1.50951 | | |

Significance levels: ***p < .001; *p < .05; CL = Collaborative Learning; SE = Self-Efficacy; SEPSA Student Engagement in Practical Skills; VB = Value-Benefits. All individual standardized regression weights are reported.

**Table 2. Pearson correlation.**

| Pearson Correlation | | | | |
|---|---|---|---|---|
| | CL | SE | VB | SEPSA |
| CL | 1 | | | |
| SE | .874** | 1 | | |
| VB | .617** | .691** | 1 | |
| SEPSA | .787** | .580** | .060 | 1 |

CL = Collaborative Learning; SE = Self-Efficacy; SEPSA Student Engagement in Practical Skills; VB = Value-Benefits.

**. Correlation is significant at the 0.01 level (2-tailed). N = 310

Table 3 shows the means scores on three different intervals (3RD week, 7th week, and 14th week). The mean scores shows the increasing trend in the student engagement in practical skill learning program developed under the scope of collaborative learning. The mean scores in the 3rd week were satisfactory which shows the effectiveness of the tasks and activities designed for the learning. The collaborative learning scheme that is designed for small teams (maximum 11 students per team) proves to be effective. Individual coaches get more interaction with the teams and teams in turn gets more interaction with the fellow students in the

**Table 3. Teams progress.**

| Teams | Tasks | | M-3RD Week | M- 7th Week | M– 14TH Week |
|---|---|---|---|---|---|
| | Offensive strategies | Defensive strategies | | | |
| 1 | Fly route | Man-to-man | 2.8158 | 4.37261 | 55349 |
| 2 | Slant route | Coverage shells (cover 2, cover 3) | 2.8158 | 5.33265 | 6.5581 |
| 3 | Out route | Zone blitz | 3.0000 | 3.62747 | 5.4419 |
| 4 | Screen Pass | Tampa 2 | 3.2368 | 4.36454 | 6.4884 |
| 5 | Button Hook | 5–5–1 Two-level defense | 3.0000 | 3.41421 | 4.3721 |
| 6 | Corner Route | Tampa 2 | 2.2105 | 3.09441 | 4.6279 |
| 7 | Hail Mary | Man-to-man | 3.2895 | 4.37365 | 5.3953 |
| 8 | Seam route | Zone blitz | 2.7368 | 3.28787 | 4.3721 |
| 9 | Fly route | Coverage shells (cover 2, cover 3) | 3.6579 | 4.51169 | 5.4186 |
| 10 | Slant route | Zone blitz | 3.5526 | 4.91293 | 5.3256 |
| 11 | Out route | 46 defense | 3.3684 | 4.00568 | 5.3023 |
| 12 | Out route | Coverage shells (cover 2, cover 3) | 3.7368 | 4.81114 | 6.2558 |
| 13 | Corner Route | Coverage shells (cover 2, cover 3) | 2.6316 | 3.10089 | 4.3953 |
| 14 | Out route | Man-to-man | 2.9474 | 3.83658 | 5.2558 |
| 15 | Fly route | Zone blitz | 2.7632 | 3.95122 | 4.1628 |
| 16 | Out route | Tampa 2 | 2.3421 | 3.77484 | 4.1860 |
| 17 | Button Hook | 46 defense | 3.6053 | 3.23655 | 4.3023 |
| 18 | Fly route | Man-to-man | 2.9474 | 3.91646 | 5.1395 |
| 19 | Corner Route | 5–5–1 Two-level defense | 2.5526 | 4.97807 | 5.2326 |
| 20 | Out route | 46 defense | 3.2368 | 3.73635 | 6.1860 |
| 21 | Seam route | Man-to-man | 2.5000 | 3.37054 | 6.2326 |
| 22 | Fly route | 5–5–1 Two-level defense | 2.7368 | 3.67145 | 5.3488 |
| 23 | Corner Route | Man-to-man | 3.0263 | 4.55071 | 6.1860 |
| 24 | Fly route | Coverage shells (cover 2, cover 3) | 2.5526 | 3.30896 | 5.3023 |

Mean scores: 1–1.9(No engagement); 2–2.9(Slight engagement); 3–3.9(Moderate engagement); 4-onward (full engagement).

group as well as with coach. This buildup student's confidence and motivation as a results in the 7$^{th}$ week the mean score gradually increased to moderate-full engagement and in the 14$^{th}$ week all students were fully motivated and engaged in practical skill learning program through collaborative learning. The increase in mean scores shows the effectiveness of the collaborative learning (CL) program with focus on student engagement in practical skill acquisition (SEPSA).

## 8. Discussion

The importance collaborative learning in student engagement has been primarily investigated in terms of its association in building students with set of practical skills through the development of self-confidence and relevance to real world opportunities. We have found that CL has a direct positive association with SEPSA and CL give opportunities to the students to increase social interaction and build self-confidence. In CL students gets chance to express their views and gets instant feedback from fellow students as well as teachers. This approach build self-confidence and students began to engage in skills acquisition. Furthermore, students assess the time and efforts put in learning skills, if they perceive the learnt skills are worth of their time and efforts they get engage in practical skill learning process. CL success not only depended upon the set of practical skills but also the worth of skills against the efforts exerted by the students in learning the set of skills. Students get motivated when the practical skills are relevant with their interest and helps them in gaining employment opportunities in near future. In CL 28 teams completed all the assigned tasks and scored well above 90+ grades no single student scored any less than 70% in an assigned tasks. Moreover, effectiveness of CL is directly dependent upon the set of skills and their practical usage the more skills are relevant with students' interest and needs of real world opportunities more will be the student engagement.

The mediating analysis further provide insight into the association that student self-confidence is important factor in student engagement and through CL students gets opportunities to develop their personalities and self-confidence. This enables them to learn better and to gain practical set of skills.

## 9. Theoretical contribution

Our study has contended the existing knowledge by exploring CL and SEPSA in depth. The concept of SE was refined with CL. CL contributes towards the development of students' personality and self-confidence. In previous studies the concept of SE was simply linked as those students who have existing high SE are capable of performing better as compared to those who lack self-confidence. We suggested that if CL tasks are developed based on student-interest and need of time this results in better learning and development of self-confidence. Additionally, Designing of CL tasks requires fair chance to all participants to get engaged in discussion. It is important to have set of learning tasks that allow equal participation and fair treatment to all students. Group size should be designed in accordance with the time-frame available for each activity and number of students. Value of each tasks needs to be linked with students efforts in terms of time and physical/mental exertion. Our study has further open up the concept of CL with SEPSA using two mediating variables that support the student learning process. Both the mediating variables are linked with psychological domain of the students and plays a key role in developing students' interest and motivation to engage in learning.

## 10. Limitation of the study

The study was conducted on sports and physical culture students studying in a first stage of Higher education at the Abai Kazakh National Pedagogical University. Due to the specific

stage of higher education and only one faculty was included in the study this put limitation on the generalization of the study. Moreover, only 310 students were included in the study this also limits the scope of the study.

## 11. Future recommendation

We recommend large and diverse sample size with longer time-lag study to better understand the importance of CL and SEPSA in terms of Smart Goals and Students' preferences. Additionally, use of virtual technologies in students' engagement can help researchers to better develop more comprehensive CL approach with practical set of skills. A comparative study between different institutes can better shed light on the concepts.

## Supporting information

**S1 File.**
(PDF)

**S1 Data.**
(XLSX)

## Author Contributions

**Conceptualization:** Yerlan Temirkhanov, Aidyn Zhumagulov, Saltanat Boztayeva.

**Data curation:** Yerlan Temirkhanov, Mira Iralina, Gulnaz Atagulova.

**Formal analysis:** Gulnaz Atagulova, Saltanat Boztayeva.

**Methodology:** Yerlan Temirkhanov, Taiyrzhan Iskakov.

**Writing – original draft:** Yerlan Temirkhanov.

**Writing – review & editing:** Yerlan Temirkhanov.

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
