## [Decision Letter · Decision Letter 0]

17 Oct 2023

Investigating the conception of collaborative learning (CL) and student engagement in the acquisition of practical skills (SEPSA) among prospective physical education and sports students.

PONE-D-23-18581

Dear Dr. Temirkhanov,

We’re pleased to inform you that your manuscript has been judged scientifically suitable for publication and will be formally accepted for publication once it meets all outstanding technical requirements.

Kind regards,

Ender Senel, PhD

Academic Editor

PLOS ONE

Journal Requirements:

Reviewers' comments:

Reviewer's Responses to Questions

**Comments to the Author**

1. Is the manuscript technically sound, and do the data support the conclusions?

Reviewer #1: Yes

Reviewer #2: Yes

2. Has the statistical analysis been performed appropriately and rigorously? 

Reviewer #1: Yes

Reviewer #2: Yes

3. Have the authors made all data underlying the findings in their manuscript fully available?

Reviewer #1: Yes

Reviewer #2: Yes

4. Is the manuscript presented in an intelligible fashion and written in standard English?

Reviewer #1: Yes

Reviewer #2: Yes

5. Review Comments to the Author

Reviewer #1: Dear author/Editor,

I recently had the opportunity to read the article titled "Investigating the Conception of Collaborative Learning (CL) and Student Engagement in the Acquisition of Practical Skills (SEPSA) among Prospective Physical Education and Sports Students," and I must say that it greatly impressed me in several ways. This review is based on the criteria outlined for publication in PLOS ONE, and I believe this article fulfills each of these criteria exceptionally well.

1. Original Research: The article unquestionably meets the requirement of presenting original research. It delves into an important topic within the field of physical education and sports, shedding light on the often-overlooked nexus between collaborative learning and student engagement in skill acquisition.

2. Publication Novelty: The statistical results are well explained and support research findings, which underscores its novelty and relevance to the field.

3. Technical Standard: The study stands out for its high technical standard. The experimental design, data collection, and statistical analyses are meticulously detailed, allowing for a thorough understanding of the methodology employed.

4. Data-Supported Conclusions: The conclusions drawn in the article are robust and are firmly supported by the presented data. The authors have done an exemplary job of connecting their findings to the research questions, leaving no room for ambiguity.

5. Clarity and Standard English: The article is written in standard English, making it accessible and intelligible to a wide audience. The clarity of the prose greatly enhances the reader's ability to comprehend the research findings and implications.

6. Ethical Standards: It is evident that the research adheres to the highest ethical standards. The authors have taken care to ensure the ethics of experimentation and research integrity, which is crucial for maintaining the trustworthiness of scientific research.

7. Data Availability: The article meets the requirement for data availability by adhering to appropriate reporting guidelines and community standards. This commitment to transparency enhances the credibility of the research.

In conclusion, the article "Investigating the Conception of Collaborative Learning (CL) and Student Engagement in the Acquisition of Practical Skills (SEPSA) among Prospective Physical Education and Sports Students" not only satisfies but excels in meeting the criteria set by PLOS ONE. It is a commendable piece of research that contributes significantly to the understanding of collaborative learning and student engagement in the context of practical skill acquisition. I wholeheartedly recommend its publication in PLOS ONE, as it undoubtedly represents a valuable addition to the scientific literature in the field of physical education and sports.

Reviewer #2: I have thoroughly reviewed the manuscript and am pleased to offer my feedback for publication. The manuscript under consideration presents a well-structured and comprehensive study on collaborative learning (CL) and student engagement in the acquisition of practical skills (SEPSA). The authors have conducted rigorous research, and their findings are both relevant and significant to the field. Furthermore, the manuscript is exceptionally well-written and organized. The introduction effectively establishes the context, and the materials and methods section provides clear and detailed information regarding the study design and data collection process. Based on the strengths of this manuscript, including its methodological rigor, clarity of presentation, original contribution, and robust data analysis, I recommend that this manuscript be accepted for publication in PLOS ONE.

6. PLOS authors have the option to publish the peer review history of their article (what does this mean?). If published, this will include your full peer review and any attached files.

Reviewer #1: **Yes: **Dr. Muhammad Ali

Reviewer #2: No

---

## [Editor Report · Acceptance letter]

19 Oct 2023

PONE-D-23-18581 

Investigating the conception of collaborative learning (CL) and student engagement in the acquisition of practical skills (SEPSA) among prospective physical education and sports students. 

Dear Dr. Temirkhanov:

I'm pleased to inform you that your manuscript has been deemed suitable for publication in PLOS ONE. Congratulations! Your manuscript is now with our production department. 

Kind regards, 

on behalf of

Dr. Ender Senel 

Academic Editor

PLOS ONE